# Work-to-Family Conflict, Job Burnout, and Project Success among Construction Professionals: The Moderating Role of Affective Commitment

**DOI:** 10.3390/ijerph17082902

**Published:** 2020-04-22

**Authors:** Jiming Cao, Cong Liu, Yubin Zhou, Kaifeng Duan

**Affiliations:** School of Economics and Management, Tongji University, Shanghai 200092, China; caojm@tongji.edu.cn (J.C.); liucong1993@tongji.edu.cn (C.L.)

**Keywords:** affective commitment, construction professionals, job burnout, project success, structural equation modeling, work-to-family conflict

## Abstract

This study explored the effects of work-to-family conflict on job burnout and project success in the construction industry. First, a theoretical model with affective commitment as a moderating variable was developed according to the conservation of resources theory. A structured questionnaire survey was then performed with Chinese construction professionals, with 309 valid responses received. In the valid data, the proportion of male construction professionals is 73% and that of female construction professionals is 27%. The analysis of the valid data used structural equation modeling. The results indicate that: (i) work-to-family conflict has a positive and significant effect on job burnout, and a negative and significant effect on project success; (ii) job burnout negatively affects project success; (iii) affective commitment negatively moderates the relationship between work-to-family conflict and job burnout. This study extends the existing body of knowledge on work-to-family conflict and helps us to better understand the functional and moderating roles of affective commitment in the context of construction projects. Furthermore, this study provides theoretical guidance and a decision-making reference to help construction enterprises manage the work-to-family conflict and job burnout of construction professionals and advance their levels of affective commitment.

## 1. Introduction

Construction is a labor-intensive and task-driven industry, characterized by high risks, heavy workloads, and long construction periods [1]. Construction is often described as a difficult and hazardous industry and construction professionals face both common, traditional challenges such as project quality, time and cost, as well as increasingly urgent safety and environmental issues [2,3]. In recent decades, construction projects have become increasingly large, complex and integrated [4]. To achieve project success, many construction enterprises encourage construction professionals to spend more time working, including evenings, weekends, and holidays [5]. These factors combined impose immense and prolonged pressure on construction professionals in implementing construction projects, possibly triggering job burnout [6]. Job burnout refers to the psychological symptoms of fatigue, cynicism, and inefficacy in the face of chronic work pressure [7]. Job burnout is considered the main reaction and product of work pressure experienced by individuals [8]. It has significant negative effects on individual-related outcomes such as wellbeing, health, and work commitment [9,10], and may affect organization-related results, such as project success. 

Compared with other industries, the construction industry involves more heavy workloads and longer average working hours [11]. This is because construction projects are high risk and very complex, and their internal and external environments are highly dynamic and uncertain [1,6]. This frequently leads to difficult tasks, complex processes, and unforeseen problems in the project implementation [12]. Furthermore, construction involves strict node planning and individual performance appraisals [13]. To complete node planning on time and achieve smooth project delivery, construction professionals must spend much time and energy at work [6]. This prevents them from effectively performing family responsibilities, such as accompanying a spouse, caring for children and elderly family members [14]. This ultimately leads to work-to-family conflict. As a form of conflict between roles, work-to-family conflict reflects the incompatibility of role stress coming from the work and family fields [15]. Work-to-family conflict has become a serious problem in the construction industry [13,14]. It negatively impacts job satisfaction, the quality of family life, and the wellbeing of construction professionals, increases their psychological pressure and triggers negative emotions, such as frustration, anxiety and anger [12,13,14,15,16]. These are likely to increase their likelihood of experiencing job burnout, affecting project success. 

To address the influence of work-to-family conflict on the job burnout of construction professionals, many researchers have focused on the alleviative effect of support, such as organizational support [6]. However, organizational support is insufficient to alleviate the effect of work-to-family conflict on job burnout in the setting of construction projects. This is because the internal and external environment of construction projects is changing and uncertain, making it difficult for construction enterprises to provide long-term and effective support [1]. Consequently, construction professionals also need to exert their own initiative to adjust their emotions and enthusiasm for work [17], such as affective commitment. Affective commitment, as the emotional bond between employees and their organizations, includes emotional attachment, identification, and employee participation [18]. Employees with high affective commitment tend to have higher passion and enthusiasm for their organizations [19], and are willing to work harder to make more in-role and extra-role contributions to realize organizational goals [20]. Therefore, affective commitment is a driving force and an important job resource [21], and may contribute to alleviating the relationship between work-to-family conflict and job burnout. 

Existing research has paid attention to the problems of work-to-family conflict in the context of construction projects [11,12,13,14]; however, few studies have investigated whether affective commitment can alleviate the relationship between work-to-family conflict and job burnout. Moreover, few studies have explored the impacts of work-to-family conflict on job burnout and project success in the construction industry. To address this research gap, this study applied the conservation of resources (COR) theory, and introduced affective commitment as a moderating variable to develop a theoretical model exploring the relationship between work-to-family conflict, job burnout, affective commitment, and project success. Structural equation modeling was applied for the empirical test. 

The objectives of this study were as follows: (i) to examine the relationship between work-to-family conflict and job burnout; (ii) to determine whether job burnout is significantly related to project success; (iii) to explore whether there is a significant relationship between work-to-family conflict and project success; (iv) to assess whether affective commitment plays a moderating role between work-to-family conflict and job burnout. This study provides theoretical and practical insights into managing work-to-family conflict and job burnout in the setting of construction projects. It also provides a reliable reference for improving the levels of the affective commitment of construction professionals. 

## 2. Theoretical Background

### 2.1. Work-to-Family Conflict 

Work-to-family conflict is a form of inter-role conflict, caused when work role requirements are incompatible with family role requirements [15]. Construction professionals have to work long hours under tremendous work pressure, due to heavy workloads, inflexible scheduling, and complex tasks and processes [1,6]. However, they have limited personal time and energy [22]. The work of construction professionals requires a lot of these limited resources, resulting in insufficient time to effectively fulfill family responsibilities [14]. Therefore, work-to-family conflict for construction professionals is inevitable. In the setting of construction projects, work-to-family conflict can be divided into time-based work-to-family conflict, strain-based work-to-family conflict, and behavior-based work-to-family conflict [16]. 

First, time-based work-to-family conflict occurs when time requirements in the field of work occupy the time from the family domain [23]. As technicians or managers of construction projects, construction professionals need to address many issues with internal and external stakeholders (e.g., owners, contractors, designers, supervisors, consultants) [24]. Moreover, construction projects are dynamic and uncertain, with difficult tasks, complex processes and unforeseen problems [6,25]. The combination of these factors causes construction professionals to work long hours under tremendous pressure, leaving insufficient time to perform family responsibilities, ultimately leading to time-based work-to-family conflict. 

Second, strain-based work-to-family conflict occurs when fatigue and irritability caused by stress in the work field limits an individual’s ability to fulfill requirements in the family field [26]. Construction professionals need to address many stressful and challenging tasks, such as taking on multiple roles and rapidly adjusting to changing project requirements [27,28]. This is likely to trigger strain-based work-to-family conflict. 

Third, behavior-based work-to-family conflicts occur when behaviors are valid in a work role, but are not effective or are even counterproductive in a family role [15]. In a dynamic and stressful project environment, construction professionals who experience heavy workloads or challenging tasks may have negative emotions, such as anger, disappointment, and frustration [29]. These may negatively affect project performance [16,29]. As such, construction professionals must maintain their emotional resilience and appropriately adjust their emotions at work [16]. At home, family members expect warmth and caring behavior [14,16]. These differences in behavioral demands between the family domain and work domain are difficult to accommodate, leading to behavior-based work-to-family conflicts. 

### 2.2. Job Burnout

Job burnout is a form of chronic emotional fatigue caused by constant and daily job stress [9]. It is generally believed that job burnout includes three dimensions: emotional exhaustion, cynicism, and low professional efficacy [30]. Emotional exhaustion involves the feeling that emotional resources are exhausted, leading to a lack of vitality. Cynicism is characterized by a cynical attitude and an exaggerated distancing from one’s work and human relationships, such as with colleagues and clients. Low professional efficacy involves professionals’ negative evaluation of themselves and their dissatisfaction with their work achievements. 

Existing studies have found that job burnout due to work stress is related to both negative individual and organizational outcomes. At an individual level, job burnout is associated with mental and physical health problems, including mental distress, depression, and sleep disorders [9,31]. At an organizational level, job burnout is associated with low levels of project commitment, reduced organization effectiveness, and an increased employee turnover rate [10]. Furthermore, existing studies have shown that job burnout is “contagious” and may spread to colleagues [32]. This results in negative emotions, such as depression, anxiety and even anger, negatively impacting family life [6]. Therefore, job burnout has negative effects at individual and organizational levels, and may also affect socioeconomic factors. 

Job burnout is considered to be the result of a conjugation of contextual factors (e.g., heavy workloads, long working hours, great pressure) and personal factors (e.g., low self-efficacy, emotional intelligence, personality traits). However, many job burnout studies have found that stress from work, organization and society in the context of work are the main predictors of job burnout in the construction industry [9,10]. In the context of construction projects, due to the temporary, one-time, and uncertain nature of construction projects, professionals face many challenging tasks and unforeseen project situations during the project implementation [6,25]. Construction professionals also have important responsibilities for the cost, duration, quality, and safety objectives of a construction project [2]. As a result, they experience significant stress from work, organization, and society over a long period, from the start of a project to delivery. This causes a high risk of job burnout. Existing research has investigated job burnout among construction professionals, and the results have shown that the level of job burnout and emotional exhaustion of construction professionals are significantly greater compared to workers in other categories who work on different projects [8,9,10,11]. Therefore, job burnout is a serious problem in the construction industry. 

### 2.3. Affective Commitment

Organizational commitment is an important topic in organizational behavior research, and refers to an individual’s attitude toward the organization [33]. There are three types of commitment: affective, normative, and continuance [34]. Affective commitment involves emotional attachment to and identification with the organization. Normative commitment arises from the perceived obligation to the organization. Continuance commitment refers to the perceived cost of leaving the organization. These three forms of commitment reflect the relationship between individuals and organizations. Among these three forms of commitment, affective commitment is believed to have the greatest impact on employees’ job outcomes [35,36]. This is because affective commitment is characterized by an attachment to the organization, acceptance of the organization’s values, and a willingness to stay in the organization [37]. Employees with high affective commitment are more motivated to achieve higher job performance and make more meaningful contributions to the organization compared to those with normative commitment or continuance commitment [35,38]. Existing studies have proved that high levels of affective commitment positively impact individual-related outcomes, including work enthusiasm, work efficiency, and job performance [34,35,36,37,38]. This indicates that affective commitment is a key predictor of employee behavior. 

Affective commitment comes from good organizational perception [38]. In recent decades, the increased complexity and uncertainty of construction projects and the increased demand for quality services have made the affective commitment of construction professionals increasingly important. Affective commitment can lead construction professionals generating positive attitudes toward their organization and commitment to a project [19]. Furthermore, construction professionals with high affective commitment are eager to remain in the organization, because they share the same values and goals with their organization and are willing to invest the effort to achieve smooth project delivery [18]. Therefore, affective commitment is one of the key factors influencing construction professionals’ behaviors. High levels of affective commitment can motivate construction professionals to improve job performance, make meaningful and positive extra-role contributions, enhance team spirit and organization cohesion, and increase organizational work efficiency [18,19,20,21]. All of these contribute to improving project performance. 

### 2.4. Project Success

Project success is a hot topic in the field of construction project management [39,40,41,42]. The existing research on project success tends to combine project governance with project management and takes the whole project life cycle into account [25]. Furthermore, project success and project management success are becoming increasingly related [39]. However, there has been no consensus on the connotations of project success. This is because different project participants have different benefit demands, resulting in different definitions of project success criteria and key factors of project success being based on different stakeholders’ perspectives (e.g., owners, contractors, supervisors, consultants, designers) [40]. For example, Shenhar et al. [41] suggested that measurements of project success include project efficiency, effects on customers, business success, and future success. Berssaneti and Carvalho [42] indicated that the project success refers to the iron triangle of time, quality, cost, and customer satisfaction. Luo et al. [4] proposed that project success is defined by time, cost, quality and safety, stakeholder satisfaction, and business value. Therefore, project success can be evaluated from the perspective of stakeholders and the inherent characteristics of construction projects. 

This study focuses on the effects of work-to-family conflict and the job burnout of construction professionals on project success. Work-to-family conflict and the job burnout of construction professionals mainly occurs during project implementation; as such, the effect of these factors on project success also occurs at this stage. With respect to construction project stakeholders, although their focus is different, the evaluation criteria for project success during the construction stage are essentially the same, and include hard and soft indicators [25]. Hard indicators include the control objectives of quality, cost, duration, and safety [4]. Soft indicators include project management effectiveness, stakeholder satisfaction, future cooperation opportunities, and the level of trust among the stakeholders [25].

The theoretical framework of this study uses the job demands–resources (JD–R) model. The JD–R model is a mainstream conceptual framework for studying work stress, work-to-family conflict and job burnout. This model describes the impact of job characteristics on job burnout. According to JD–R model, job characteristics can be classified into two categories: job demands and job resources. Job demands refer to the social or organizational requirements involved in an individual’s work that require continuous energy, time or skills, which, in turn, are related to certain physical and psychological costs, such as work pressure and work-to-family conflicts. Job resources refer to material, psychological or organizational resources that help individuals achieve their work goals. According to the JD–R model, researchers can better understand and predict individual job burnout. In the construction industry, long working hours and heavy tasks reflect the job demands of construction professionals. These high-load job demands will result in work-to-family conflicts, thus putting construction professionals at high risk of job burnout. Therefore, this study uses the JD–R model for theoretical analysis and develops research hypotheses to investigate the relationship between work-to-family conflict, job burnout and project success among construction professionals. 

## 3. Hypotheses Development and Theoretical Model 

### 3.1. Hypotheses Development

#### 3.1.1. Work-to-Family Conflict and Job Burnout 

Work-to-family conflict is a source of pressure that makes employees unable to effectively fulfill family responsibilities [6]. This can have negative impacts, such as low job satisfaction and high turnover rates [15]. According to the conservation of resources (COR) theory, people work to acquire and maintain resources that contribute to their objectives, such as improving the quality of their life and increasing family wellbeing [43]. However, when someone is unable to effectively perform their family responsibilities, there is likely to be the potential or actual loss of resources, such as a decreased sense of wellbeing, poor quality marital relationships, and even divorce [16]. The threat of resource loss is a major cause of pressure and is likely to trigger job burnout [44]. In this case, people may address the pressure by taking measures such as quitting a job to minimize the loss of resources and to protect their remaining resources [6]. Anderson et al. [45] pointed out that work-to-family conflict leads to low work satisfaction and high turnover intention, which may lead to job burnout. Lambert et al. [46] found that work-to-family conflict has a positive relationship with job burnout among correctional staff. Wang et al. [47] revealed that work-to-family conflict has a positive effect on job burnout among female nurses. 

In the context of construction projects, heavy workloads, inflexible scheduling, changing requirements and complex tasks and processes make construction professionals unable to effectively fulfill their family responsibilities [1,6]. This leads to work-to-family conflict, resulting in lost energy and personal time, ultimately causing job burnout. The direct result of job burnout is that construction professionals leave their organization and find another appropriate job to better balance the requirements of the work and family domains [6]. Moreover, the loss of resources (e.g., wellbeing, personal time, and energy) can lead construction professionals to produce negative emotions [48]. These negatively affect their physical and mental health. Therefore, the following hypothesis was proposed: 

**Hypothesis 1** **(H1).**
*Work-to-Family Conflict Positively Influences Job Burnout.*


#### 3.1.2. Job Burnout and Project Success

Existing studies have found that job burnout is closely linked to negative outcomes for individuals and organizations. At an organization level, job burnout has been related to reduced organizational productivity, increased employee turnover rate, and reduced organizational efficiency [9,31]. For individuals, job burnout has been closely related to low levels of job satisfaction, decreased career commitment, and increased intended turnover [10]. These negative outcomes can bring large costs to both the individual and organization. Moreover, job burnout is “contagious”, meaning that the job burnout of individuals can negatively affect their colleagues [32]. 

The construction industry is task-oriented, and the ultimate objective of a construction enterprise is to realize project success by controlling project costs, improving project quality, ensuring safe project production, and completing projects within a specified time period [49]. Construction projects have long construction periods, strict project planning, and dynamic internal and external settings [13]. As such, construction professionals need to work long hours under great pressure, which tends to cause job burnout. Negative emotions, such as anger, disappointment, and frustration caused by job burnout increase the possibility of safety accidents during project implementation [6]. Negative work attitudes, such as low professional commitment and job satisfaction caused by job burnout, can reduce the work efficiency and effectiveness of construction professionals [50]. This could cause the project to not be completed on schedule, ultimately negatively affecting project success. Therefore, the following hypothesis was proposed:

**Hypothesis 2** **(H2).**
*Job Burnout Negatively Influences Project Success.*


#### 3.1.3. Work-to-Family Conflict and Project Success

Work-to-family conflict can lead to negative results, such as decreased job satisfaction, increased psychological pressure, decreased quality of family life, and poor marital relationships [12,13,14]. This is because resources, such as family wellbeing, are lost when there is work-to-family conflict [51]. The COR theory proposes that when resources reach the lowest acceptable level, employees stop working hard to preserve personal resources and accept a decline in job performance [52]. 

Construction projects are featured by complexity and uncertainty [25]. Construction professionals must devote much energy and time to work. As such, they cannot effectively perform their family responsibilities, such as raising children, accompanying spouses, and caring for the elderly [14]. This ultimately leads to work-to-family conflict. This form of conflict is often accompanied by the loss of large amounts of energy and personal time, which can lead to bad emotions, such as anxiety, anger, and frustration [53]. It can also generate negative work attitudes, such as low project commitment, ultimately resulting in low work efficiency and performance [15,16]. Furthermore, work-to-family conflict can significantly increase the turnover intention of construction professionals [54]. When considering the key influencing factors of project success, construction professionals directly influence the completion of project objectives such as quality, duration, cost, and safety, and collaboration among project participants during project implementation [6]. The higher their turnover intention, the lower the team spirit and organizational cohesion, leading to low organizational efficiency [54]. These factors do not support the achievement of project success. Therefore, the following hypothesis was proposed: 

**Hypothesis 3** **(H3).**
*Work-to-Family Conflict Negatively Influences Project Success.*


#### 3.1.4. The Moderating Effects of Affective Commitment 

Work-to-family conflict is a source of role pressure, and job burnout is considered a direct pressure response [6,8]. Thus, there is a strong pressure–strain relationship between work-to-family conflict and job burnout. However, affective commitment may alleviate this relationship, because it can reduce the urge to save resources by employees who experience work-to-family conflicts. Affective commitment reflects the relationship between employees and their organizations [18]. Employees with affective commitment are characterized by an attachment to their organizations, their recognition of the objectives of the organization, pride in their organizations, and their strong desire to remain in the organization [36]. Therefore, employees with affective commitment have a strong sense of ownership and regard the interests of the organization as their own. They are also willing to invest more effort in order to realize organizational objectives, even if many required actions go beyond their role responsibilities [55]. According to the COR theory, individuals who are highly loyal and attached to the organization do not think it is a frustrating thing to work long hours or have constant work pressure. They also do not reduce their job satisfaction and conserve their own resources, such as time and energy, because of the high workload. Instead, they are more willing to devote their time and energy to contribute to the organization. 

Construction professionals work long and irregular hours under immerse stress, ultimately leading to work-to-family conflict [16]. Construction professionals who are stressed due to work interfering with family life tend to attribute such pressure to high-intensity work [56]. This can result in low job satisfaction and trigger job burnout. In this case, construction professionals with affective commitment can modify the attribution level and remain satisfied [34]. Furthermore, they are more likely to reduce the urge to save resources, thereby reducing job burnout [20,21]. As a result, the present study highlights that affective commitment can, to some extent, reduce the job burnout caused by work-to-family conflict. Hence, the following hypothesis was proposed: 

**Hypothesis 4** **(H4).**
*Affective Commitment can Negatively Moderate the Relationship between Work-to-Family Conflict and Job Burnout.*


### 3.2. Theoretical Model

Work-to-family conflict can lead to job burnout for construction professionals, because they feel potential or actual resource loss [6]. To minimize the loss of resources and protect remaining resources, they may take actions to exit [51]. This negatively affects the completion of project tasks, which could ultimately undermine project success. However, affective commitment may alleviate the relationship between work-to-family conflict and job burnout because individuals with affective commitment can modify their stress levels and reduce the urge to save resources. Therefore, there may be a close relationship between work-to-family conflict, job burnout, affective commitment, and project success. According to the proposed hypotheses and combined with construction project characteristics, this study introduced affective commitment as a moderating variable to develop the theoretical model shown in Figure 1. 

## 4. Method 

### 4.1. Questionnaire Design

To test the theoretical model, a questionnaire (Appendix A) was designed to measure the studied variables. Basic demographic data such as family information were also investigated. Study variables included work-to-family conflict, job burnout, affective commitment, and project success. Three steps were applied to develop the questionnaire’s measurement scale. The first step was to identify measurement items in the existing literature shown to have a high-level of reliability and validity [57]. The original scales were developed in English; as such, all items were back-translated and modified [58]. The second step was to modify and improve the existing measurement items according to the characteristics of Chinese construction projects [14]. The third step was to optimize the measurement items by conducting on-site discussions with experts in construction project management [25]. 

All items were designed based on relevant previous studies [1,6]. The selection criteria of the measurement items are determined according to the test results of the reliability and validity of the items in each variable measurement scale in previous studies. The reliability coefficients of corrected-item total correlation (CITC) and Cronbach’s α were used to explore the reliability and validity of the items in each variable measurement scale [4]. CITC reflected the reliability of the items. Items with CITC values greater than 0.5 were selected [25]. Cronbach’s α reflected the validity of the items, which should not be below 0.7. 

Work-to-family conflict was measured using six items; job burnout was measured using eight items; affective commitment was measured using six items; project success was measured using eight items. The measurement model used in this study provided the relationships between work-to-family conflict, affective commitment, job burnout, and project success (latent variables), and their respective groupings (observable variables) [59]. Furthermore, this study was consistent with the reflection model, because each observable variable in the measurements reflected the latent variables [60]. The relationship directions flowed from the latent variables to the observable variables [61]. Eventually, the items used to measure work-to-family conflict were designed according to previous studies [14,62,63]. The items used to measure job burnout were designed with reference to the relevant literature [64,65]. The items used to measure affective commitment were also designed according to previous studies [37,66]. The items used to measure project success were designed with reference to the relevant literature [25,67,68]. 

Face-to-face interviews with experts in the construction field were used to optimize the developed measurement items, ensuring their suitability [4]. Representatives of owners, contractors, supervisors, consultants and designers were interviewed to collect their professional opinions on the suitability of the measurement items of the four key variables (work-to-family conflict, affective commitment, job burnout and project success). A total of nine experts from different project teams, who served as project managers, department managers, professional managers, and project engineers were selected. After two rounds of face-to-face discussion, we reached a consensus on the appropriateness of the measurement items (Table 1). All measurements were evaluated using a five-point Likert scale (i.e., one means ‘strongly disagree’, five means ‘strongly agree’). 

### 4.2. Pilot Test Procedure

The goal of the pilot test was to validate and modify the initial questionnaire [69]. The pilot test was conducted in construction projects in Shanghai, Zhejiang Province, and Jiangsu Province in China. Respondents mainly included technicians and middle-level and high-level managers from owner teams, contractor teams, supervisor teams, designer teams, and consultant teams. A total of 390 questionnaires were distributed by email and express delivery. Of the 163 returned questionnaires, 107 questionnaires were deemed valid, which is a response rate of 27% (107/390). Questionnaire purification included the removal of any questionnaire where: (1) the answers were clearly not serious, (2) items were unanswered or (3) answers contradicted each other [67]. 

Before the pilot test, we implemented a normality test on the valid data. Specifically, a normal quantile–quantile plot (Q–Q plot) was used to examine whether the valid data aligned with the normal distribution; the Q–Q plot is the most commonly used and effective diagnostic tool for testing normal distribution [70]. Figure 2 shows the normal distribution of the valid data. It shows that the sample distribution of different variables, including work-to-family conflict, affective commitment, job burnout, and project success, is almost a straight line. Therefore, the valid data satisfied the normal distribution and could be tested further. 

The pilot test included three steps. First, the reliability coefficient of corrected-item total correlation (CITC) and Cronbach’s α were used to purify all measurement items [4]. The CITC reflects the reliability of the items. When the CITC value was below 0.5, the item was deleted [25]. Cronbach’s α was used to test internal consistency, which should not be below 0.7 [71]. The greater the Cronbach’s α, the stronger the correlation between the items and internal consistency. Second, the Kaiser–Meyer–Olkin (KMO) test and Bartlett test were used to evaluate whether exploratory factor analysis could be performed [72]; it was determined that an exploratory factor analysis should be implemented for variables with a KMO value above 0.6 [73]. The third step was the exploratory factor analysis. These three steps confirmed both valid and invalid measurement items. By purifying all measurement items, a formal questionnaire with large-scale sampling could be administered. 

### 4.3. Participants and Formal Data Collection

There was no sampling framework for this survey; as such, a non-probability sampling method was adopted to obtain a representative sample [25]. This approach was determined to be appropriate, because the non-probability sampling is more suitable for this in-depth quantitative study, in which we focus on a complex and professional phenomenon in the construction industry. In addition, the respondents were not randomly chosen from the population, but were chosen from professionals in the construction industry and based on their willingness to participate in the study [74]. These measures ensure that the respondents are practitioners in the construction industry and are willing to provide the required information in the questionnaire. The survey samples were collected from construction professionals in Shanghai, Zhejiang Province, and Jiangsu Province in China, and included technicians and middle-level and high-level managers of the owner teams, contractor teams, supervisor teams, designer teams, and consultant teams from different construction projects. A total of 1200 questionnaires were distributed using email and express delivery; 386 questionnaires were returned. After questionnaire screening, 309 were determined to be valid, with a response rate of 26% (309/1200). This was considered normal, based on a 20%–30% response rate in most construction industry studies [75]. These valid data were used for reliability and validity analysis and structural equation modeling analysis. Before these analyses, we also applied a Q–Q plot to perform a normality test of the valid data. The results showed that these valid data aligned with a normal distribution. Additionally, the sample structure of valid questionnaires was analyzed; Table 2 shows the categories and levels. 

## 5. Results of Confirmatory Factor Analysis

The confirmatory factor analysis of work-to-family conflict, affective commitment, job burnout, and project success was implemented using AMOS 21.0, a professional structural equation modeling program [25]. Item reliability indicators and the factors of construct reliability (CR) were obtained through the analysis. Measurement items with standardized factor loadings below 0.6 were removed [76]. CR was used to assess consistency among the measurement items. Good structural reliability requires a CR value that exceeds 0.6 [25]. Convergence validity was tested by the average variance extracted (AVE). A good convergence validity of the variable measurement items requires an AVE value greater than 0.5 [77]. Fitting indicators were used to evaluate the goodness-of-fit, and included the ratio of the chi-square statistic to the degrees of freedom ($x^{2}/\mathrm{df}\;$), root mean square error of approximation (RMSEA), goodness-of-fit index (GFI), comparative fit index (CFI), adjusted goodness-of-fit index (AGFI), incremental fit index (IFI), and the normed fit index (NFI) [78]. 

All indicators of research variables met the demands, and the standardized factor loadings of all measurement items were greater than 0.6. Each potential variable’s CR value exceeded 0.7, indicating a high overall reliability and internal consistency of the measurement items. The AVE value of each potential variable exceeded 0.6, which indicates good convergence validity. Table 3 shows the specific results of the confirmatory factor analysis. In addition, non-response biases and common method bias were examined using the chi-square method and Harman’s single-factor test, respectively [79]. The results suggested that the significant heterogeneity among the variables and the common method bias was not a significant concern, signaling the ability to move forward with the theoretical model [78]. 

## 6. Model Testing and Results

Before testing the theoretical model, we considered specific demographic variables, including gender and marital status, as these may affect job burnout and project success [80]. Levene’s test of homogeneity of variances was considered to be the appropriate method to test for cross-group mean differences. This is because Levene’s test of homogeneity of variances is mainly used to test whether the variances of two or more groups of samples are homogeneous, and this test method only requires the samples to be random. However, other common test methods, such as Bartlett’s test of homogeneity of variances, are mainly used to test the data subject in terms of normal distribution, and its test results for the data subject regarding the deviation of normal distribution are inaccurate. Levene’s test can be used to test the data subject in terms of normal distribution or the data subject in terms of unclear distribution. Besides, the three conversion forms of data in Levene’s test principle ensure the robustness and accuracy of the results. 

SPSS 23.0 was used to examine the influence of these control variables on dependent variables. The results indicate that the effect of gender and marital status on the project success is not significant (gender, −0.127, *p* > 0.05; marital status, 0.043, *p* > 0.05). In addition, considering that the work experience of employees and their job position may impact project success [1,81], this study considered these variables in order to complement the related research. The results show that employees’ work experience and job position have no significant impact on project success (work experience, 0.184, *p* > 0.05; job position, 0.059, *p* > 0.05). Older and younger construction professionals may respond differently to work-to-family conflict and job burnout [1]. As such, a homogeneity of variance test was performed to assess work-to-family conflict (Levene statistic = 0.218, *p* > 0.05) and job burnout (Levene statistic = 0.103, *p* > 0.05). The results show that the homogeneity of variance hypothesis is effective, indicating that older and younger construction professionals respond similarly to work-to-family conflict and job burnout. 

Structural equation modeling was used to verify the theoretical model; this form of modeling is considered an appropriate tool for exploring the relationships among variables and has been widely used in research in the construction field [78]. The analysis of structural equation modeling was conducted with AMOS 21.0. Figure 3 and Table 4 show the results and indicate that the fitting index meets the demands. Specifically, $x^{2}/\mathrm{df}$ is 2.37, which is lower than the strict limit of three [25]. The RMSEA is 0.067, which is lower than 0.08, indicating a good fit [78]. The NFI, IFI, GFI, and AGFI are 0.92, 0.95, 0.93, and 0.91, respectively. All these values exceeded the threshold of 0.9 [82]. 

Table 4 shows that all hypotheses passed the tests for significance. Firstly, the effects of work-to-family conflict on job burnout are positive and significant (work-to-family conflict→job burnout, 0.406, *p* < 0.001). This supports H1. Secondly, the effects of job burnout on project success are negative and significant (job burnout→project success, −0.347, *p* < 0.001). This supports H2. Thirdly, the effects of work-to-family conflict on project success are negative and significant (work-to-family conflict→project success, −0.129, *p* < 0.01). This supports H3. Finally, affective commitment negatively moderates the relationship between work-to-family conflict and job burnout (work-to-family conflict * affective commitment→job burnout, −0.132, *p* < 0.05). This provides support for H4, indicating that affective commitment can alleviate the negative effects of work-to-family conflict on job burnout. 

This study formed the cross-product terms of the indicators when verifying the moderating effect of affective commitment. The approach we adopted was cross-product terms analysis, which included a mean structural analysis. A simple slope analysis was also implemented to better understand the moderating effect of the interactions between terms [1]. Figure 4 shows the slope. Non-parallel lines represent the existence of moderating effects. The green, blue, and red lines present the moderator’s high, medium, and low conditions, respectively. The moderation analysis indicates that affective commitment negatively moderates the relationship between work-to-family conflict and job burnout, which further verifies the moderating role of affective commitment. 

## 7. Discussion

### 7.1. Effects of Work-to-Family Conflict on Job Burnout

The results show a positive correlation between work-to-family conflict and job burnout. These results support the conclusions of Lingard et al. [64], and further verify that work-to-family conflict also aggravates job burnout in the setting of construction projects. The construction industry is featured by heavy workloads, long and irregular working hours, and inflexible scheduling [1,6]. This gives construction professionals little time to participate in family activities. Furthermore, the high complexity and uncertainty of construction projects and changing projects requirements bring tremendous pressure to construction professionals [25]. This means construction professionals cannot effectively fulfill family responsibilities. However, most construction professionals are not young or single. In addition to being responsible for the quality, duration, cost, and safety objectives of the construction project, they also have the responsibilities of raising children, accompanying spouses, and caring for elderly family members [14,16]. An inability to effectively perform family responsibilities will cause work-to-family conflicts, ultimately triggering job burnout. This manifests behaviorally as an unwillingness to face daily work, a gradual loss of interest in the work, doubts about the meaning of the work, and a feeling of collapse. Moreover, in many cases, construction professionals receive extra work tasks by phone or email, even when they are at home [6]. This extra work does not produce high work efficiency and job performance [83], but rather results in low job satisfaction and wellbeing and an increase in job burnout. 

### 7.2. Effects of Job Burnout on Project Success 

The study results show that the job burnout of construction professionals negatively affects project success, further validating research conclusions provided by Wu et al. [1,6]. There are many interdependent tasks and procedures in the implementation of construction projects [25]. This requires construction professionals to have good interpersonal skills, to ensure they can work together well and complete tasks on time. However, job burnout can cause negative attitudes of construction professionals towards work, including decreased job satisfaction and reduced project commitment [10]. It can also lead to negative behaviors, such as disrespect, distrust, and disgust among colleagues [9,31]. Therefore, job burnout can lead to low team spirit and organizational cohesion, and can undermine the relationship and trust among construction professionals and the relationships and cooperation among project stakeholders. These will directly affect the schedule and cost of construction projects. Moreover, construction professionals who experience job burnout tend to have low job satisfaction and wellbeing. As a result, they may exhibit a range of withdrawal behaviors, such as absenteeism and quitting their jobs [6]. All these negative attitudes and behaviors created by job burnout will ultimately negatively affect project success.

### 7.3. Effects of Work-to-Family Conflict on Project Success 

The results show that work-to-family conflict negatively affects project success. This finding is consistent with the finding of An et al. [84] and Hao et al. [85], who found a significant negative relationship between work-to-family conflict and job outcomes in China. However, this finding is inconsistent with the conclusion of Allen et al. [86], who found no significant relationship between work-to-family conflict and job outcome in America. This may be because, as China’s economic and social development and the quality of people’s lives gradually improve, the pursuit of people has shifted from simply pursuing a higher income to focusing on individual health and high-quality professional welfare [14]. High-quality professional welfare is closely and significantly related to individual wellbeing and job satisfaction [6]. In the setting of construction projects, due to the characteristics of long and irregular working hours, constant changing project requirements, and complex tasks and processes, construction professionals must invest significant time and energy into work, leading to work-to-family conflict [1,14]. The significant psychological pressure brought by work-to-family conflict leads to negative emotions. These are likely to cause a low level of job satisfaction and wellbeing, ultimately increasing turnover. The rapid economic development of China has provided construction professionals with more employment opportunities and higher prospects for rotation than ever before [6]. As a result, many construction professionals quit their jobs to seek another one with improved professional welfare. These negatively impact the completion of project tasks, ultimately negatively affecting project success. 

### 7.4. The Moderating Role of Affective Commitment

The results show that the affective commitment of construction professionals negatively moderates the relationship between work-to-family conflict and job burnout. This means that the higher the affective commitment of construction professionals is, the weaker the negative impacts of work-to-family conflict on job burnout are. This finding extends our understanding of the functional role and moderating role of affective commitment in the setting of construction projects. It also complements the existing body of knowledge associated with work-to-family conflict in the construction domain, by exploring how affective commitment alleviates the relationship between work-to-family conflict and job burnout. 

Following the arguments of self-justification, construction professionals who suffer from great psychological stress from work-to-family conflict tend to attribute this pressure to high-intensity work [56]. This results in their low job satisfaction and eventually triggers job burnout. However, affective commitment can alleviate this pressure–strain relationship between work-to-family conflict and job burnout. As a driving force, affective commitment is expressed by employees’ emotional attachment to their organization, adherence to the organization’s values and objectives, and a strong willingness to invest effort to achieve the organization’s goals [36]. Therefore, construction professionals with affective commitment can modify their level of attribution and remain satisfied. According to the COR theory, employees who are attached to the organization, proud of the organization, and identified with the organization’s goals and values will not be frustrated and anxious about the continuous pressure and high load at work. They are more willing to contribute their own resources such as time and energy to the organization, rather than conserve their own resources. Therefore, employees with affective commitment will not reduce job satisfaction due to continuous pressure and heavy tasks at work, which will reduce their likelihood of experiencing job burnout.

## 8. Conclusions and Implications

### 8.1. Conclusions

This study applied a structural equation model to empirically analyze and discuss the relationship between work-to-family conflict, job burnout, and project success for construction professionals in China, with a focus on the moderating role of affective commitment. The results indicate that: (i) work-to-family conflict has a positive impact on job burnout; (ii) job burnout is negatively related to project success; (iii) work-to-family conflict negatively affects project success; (iv) affective commitment negatively moderates the relationship between work-to-family conflict and job burnout. These findings advance our understanding of the dysfunctional role of work-to-family conflict and job burnout, and the functional role of affective commitment in the context of construction projects. Furthermore, this study expands the existing body of knowledge on work-to-family conflict in the field of construction project management. These findings could help construction enterprises and other project-oriented organizations better manage their employees’ work–family interactions and enhance their affective commitment, thereby advancing project success. 

### 8.2. Theoretical Implications 

Few studies have explored the relationship between work-to-family conflict, job burnout, and project success in the field of construction project management, including introducing affective commitment as a moderating variable. This study contributes to the understanding of work-to-family conflict, job burnout, affective commitment, and project success by linking these four concepts in the context of construction projects. Firstly, the study extends the existing body of knowledge on work-to-family conflict by researching work-to-family conflict as an antecedent and affective commitment as a moderating variable between work-to-family conflict and job burnout. The results validate the dysfunctional effects of work-to-family conflict and capture the interaction process between work-to-family conflict, job burnout, and project success in the context of construction projects. Moreover, this study indicates the importance of considering socio-economic factors when discussing the different consequences of work-to-family conflict. This also advances our understanding of the role of socio-economic factors in work-to-family conflict studies.

This study also explores the potential impact of affective commitment on the relationship between work-to-family conflict and job burnout. The results reveal the functional role and moderating role of affective commitment. Construction projects are dynamic, uncertain, and complex. The affective commitment of construction professionals is critical for project success, so it deserves more attention. Furthermore, although project success has been emphasized as an important topic when researching construction project management, only a few studies have explored the relationship between individual-related antecedents and project success. This study supplements those related studies by establishing work-to-family conflict as an antecedent variable.

### 8.3. Practical Implications 

This study’s theoretical model and findings have practical significance for both construction enterprises and construction professionals. First, construction professionals experience high work-to-family conflicts due to long working hours, heavy workloads, and inflexible scheduling. This can undermine their enthusiasm and passion to their work and organization, and cause negative emotions and job burnout, ultimately negatively affecting project success. Therefore, construction enterprises should focus on the work-to-family conflict and job burnout issues of construction professionals and strive to become family-friendly organizations and to create a family-supported organizational atmosphere, instead of just focusing on organizational productivity. Construction enterprises should consider optimizing work plans to ensure that construction professionals have enough personal time to perform their family duties, and give them enough time to be with their families before they participate in the next project. This would reduce their work-to-family conflict and the resulting job burnout. Additionally, if employees must work overtime, construction companies should provide them with appropriate compensation, such as bonuses and family-related holidays. 

Secondly, construction enterprises should pay attention to the functional role of affective commitment and strengthen the cultivation of construction professionals’ affective commitment. This should enhance employees’ identification and emotional attachment to their organization, improving work enthusiasm, work efficiency, and performance, and ultimately advancing project success. Measures to enhance the affective commitment of construction professionals should be considered, such as giving construction professionals sufficient decision-making opportunities at work, providing sufficient development opportunities, and developing fair incentive mechanisms. Thirdly, good communication between construction professionals and their organization is important [25], because two-way communication helps organizations better understand the specific difficulties of construction professionals regarding their work and families, and provide them with support and assistance. These measures should reduce work-to-family conflict and the resulting job burnout among construction professionals, and strengthen their affective commitment. This should improve work passion and performance, eventually advancing project success. 

### 8.4. Limitations and Future Work

Research on work-to-family conflict has been ongoing for decades. However, there has been little research on work-to-family conflict in the field of construction project management. Moreover, few studies have explored the relationship between work-to-family conflict, job burnout, and project success in the setting of construction projects. This study fills this gap, by introducing affective commitment as a moderating variable to verify a new theoretical model. This study verifies previous research conclusions, and identifies its own meaningful findings. 

However, like all studies, this research has some limitations. Firstly, the sample data came from specific regions in China. One recommended future direction is to collect data from different countries or regions, to further investigate the relationship between work-to-family conflict, job burnout, affective commitment and project success from different cultural perspectives. Secondly, this study only applied one variable, affective commitment, as a moderator variable to explore the impact of work-to-family conflict on job burnout. Future research should consider other moderating variables, such as perceptions of fairness and the organizational climate. Thirdly, an employee’s affective commitment may become more complex under some specific circumstances. Therefore, future research should explore the driving mechanism of affective commitment. Finally, this study did not consider the effect of personality traits, such as the big five personality traits. Future studies could integrate personality traits into the study, and investigate the reactions of construction professionals to work-to-family conflict and job burnout under the influence of different traits.

Even with these limitations, this study provides a useful reference for a construction enterprise to develop effective strategies to manage work-to-family conflict and job burnout and to advance affective commitment, thereby contributing to project success. 

## Figures and Tables

**Figure 1 ijerph-17-02902-f001:**
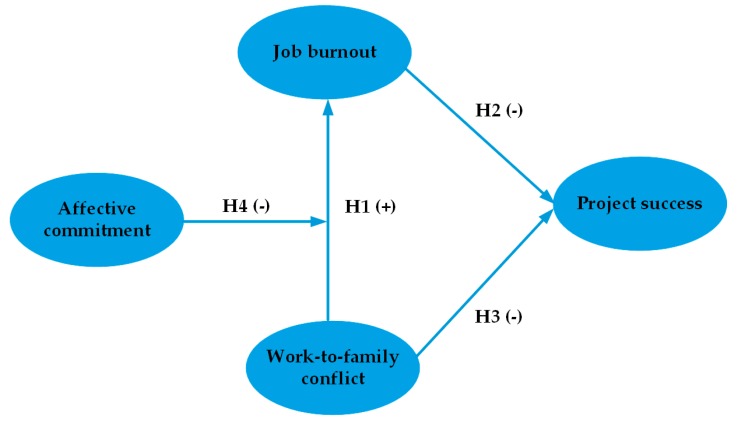
Theoretical model.

**Figure 2 ijerph-17-02902-f002:**
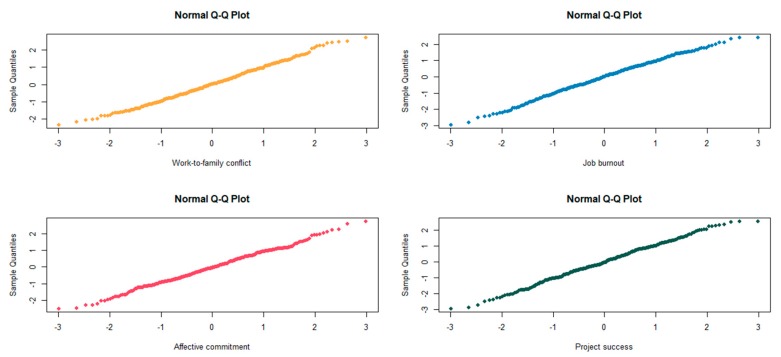
Quantile–quantile (Q–Q) plots of variables for the sample.

**Figure 3 ijerph-17-02902-f003:**
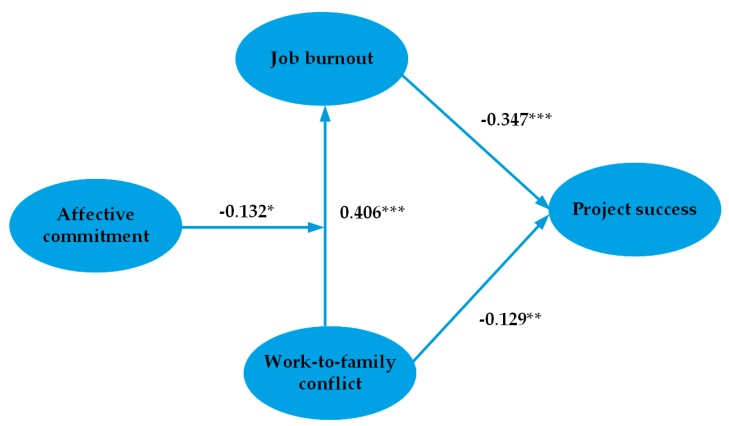
Structural equation modeling (SEM) test results of the theoretical model. Note: *, *p* < 0.05. **, *p* < 0.01. ***, *p* < 0.001.

**Figure 4 ijerph-17-02902-f004:**
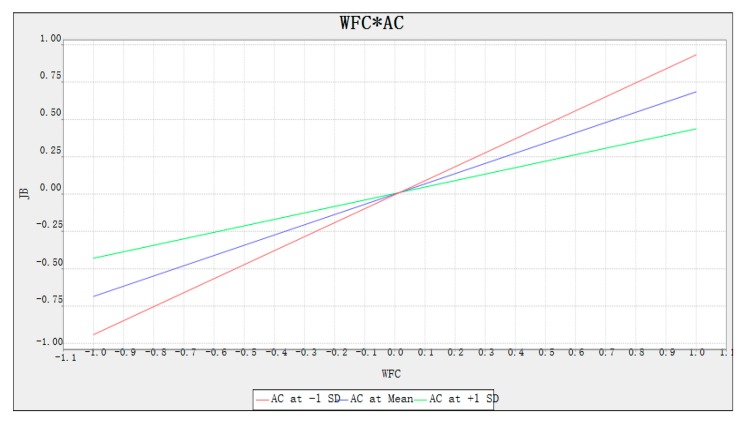
Moderation of affective commitment on the relationship between work-to-family conflict and job burnout. Note: work-to-family conflict (WFC), job burnout (JB), affective commitment (AC).

**Table 1 ijerph-17-02902-t001:** Measurements for variables.

Variables	No.	Measurement	References
Work-to-family conflict (WFC)	WFC1	The requirements of work interfere with my family life.	[14,62,63]
	WFC2	My work takes too long, so it is difficult for me to perform my family duties.	
	WFC3	Because of the demands of my work, what I want to do at home cannot be done.	
	WFC4	The pressure of work makes it difficult for me to perform my family duties.	
	WFC5	I have no interest in taking part in family activities after work.	
	WFC6	Because of my work duties, I have to change my plans for family activities.	
Job Burnout (JB)	JB1	Work makes me physically and mentally exhausted.	[64,65]
	JB2	I feel exhausted after work.	
	JB3	When I wake up in the morning and have to face the work of the day, I feel very tired.	
	JB4	It’s stressful for me to work all day.	
	JB5	Work makes me feel like I am falling apart.	
	JB6	Since I started this job, I have become less and less interested in my job.	
	JB7	I’m not as passionate about my work and my colleagues as before.	
	JB8	I doubt the significance of my work.	
Affective commitment (AC)	AC1	My values are similar to those of the construction enterprise where I work.	[37,66]
	AC2	I am concerned about the future of the construction enterprise where I work.	
	AC3	I am proud to tell other people that I work in this construction enterprise.	
	AC4	Achieving project goals is as important to me as it is to the project.	
	AC5	I am willing to work harder than ever to help this construction enterprise make progress.	
	AC6	For me, this is the best of all possible construction enterprises for which to work.	
Project Success (PS)	PS1	The progress of this project is on schedule.	[25,67,68]
	PS2	This project is within budget.	
	PS3	The project passed the acceptance check and was successfully delivered.	
	PS4	Most problems encountered in the implementation of the project can be addressed.	
	PS5	The project process is satisfactory.	
	PS6	The owner is satisfied with the project results.	
	PS7	The project meets the special requirements of client.	
	PS8	We look forward to cooperating with the other party again in the future.	

**Table 2 ijerph-17-02902-t002:** Demographic characteristics of respondents.

Characteristic	Category	Frequency	%
Gender	Male	226	73.14
	Female	83	26.86
Age	<30	53	17.15
	30–39	134	43.37
	40–50	89	28.80
	>50	33	10.68
Marital status	Single	73	23.62
	Married	236	76.38
Dependent children (aged 18 years or below)	Yes	213	68.93
	No	96	31.07
Elderly dependents	Yes	237	76.70
	No	72	23.30
Work experience	<5 years	94	30.42
	6–10 years	113	36.57
	11–15 years	46	14.89
	16–20 years	32	10.36
	>20 years	24	7.76
Job position	Project manager	31	10.32
	Department manager	67	21.68
	Project engineer	103	33.33
	Professional manager	91	29.45
	Others	17	5.22
Average hours worked per week	<40 h	19	6.15
	41–50 h	42	13.59
	51–60 h	143	46.28
	>60 h	105	33.98

**Table 3 ijerph-17-02902-t003:** Results of confirmatory factor analysis.

Variables	CR	AVE	Fit indices
Work-to-family conflict	0.87	0.79	$\;x^{2}/\mathrm{df}=2.74$; RMSEA = 0.078; GFI = 0.94; AGFI = 0.93; NFI = 0.95; IFI = 0.90
Job burnout	0.84	0.74	$\;x^{2}/\mathrm{df}=1.97$; RMSEA = 0.071; GFI = 0.95; AGFI = 0.92; NFI = 0.94; IFI = 0.91
Project success	0.81	0.71	$\;x^{2}/\mathrm{df}=1.68$; RMSEA=0.060; GFI = 0.93; AGFI = 0.91; NFI = 0.95; IFI = 0.94
Affective commitment	0.79	0.67	$\;x^{2}/\mathrm{df}=1.73$; RMSEA = 0.067; GFI = 0.94; AGFI = 0.90; NFI = 0.91; IFI = 0.92
All variables	0.76	0.65	$\;x^{2}/\mathrm{df}=1.93$; RMSEA = 0.063; GFI = 0.91; AGFI = 0.93; NFI = 0.91; IFI = 0.92

**Table 4 ijerph-17-02902-t004:** Results of theoretical model analysis.

Hypothesis	Path Coefficient	C.R. Values	S.E. Values	T Statistics	*p* Values	Hypotheses Decision
WFC→JB	0.406 ***	12.124	0.193	6.624	0.000	H1: Supported
JB→PS	0.347 ***	−4.986	0.176	−5.428	0.000	H2: Supported
WFC→PS	−0.129 **	−3.286	0.105	−3.246	0.001	H3: Supported
WFC×AC→JB	−0.132 *	−2.526	0.101	−2.513	0.014	H4: Supported
Fit indices (the full model)	$\;x^{2}/\mathrm{df}=2.37$; RMSEA = 0.067; GFI = 0.93; AGFI = 0.91; NFI = 0.92; IFI = 0.95					

Note: work-to-family conflict (WFC), job burnout (JB), affective commitment (AC), project success (Ps), critical ratio (C.R.), standard error (S.E.), *, *p* < 0.05. **, *p* < 0.01. ***, *p* < 0.001.

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
