# Peer review of "Work-to-Family Conflict, Job Burnout, and Project Success among Construction Professionals: The Moderating Role of Affective Commitment"

_ijerph, 2020, doi:10.3390/ijerph17082902_

Round 1
Reviewer 1 Report
The article shows a theme that is important today. The economic and health complications resulting from the burnout are undeniable. To this we must also add other variables associated with the reconciliation of family and work life that make the problem even more multidimensional. For this reason, it is necessary to continue investigating this issue. This article enjoys wide literary and scientific support. The following are a number of observations that may be useful. Thank you very much.
- With respect to the abstract, it would be interesting to introduce the percentage of women or men to get an idea of the representation they have.
- The key words could be in alphabetical order and one of them should be related to the research design.
- Both the introduction and the theoretical framework deal in depth with the subject. The hypotheses are highly developed.
- Throughout the manuscript, sometimes the spacing is suddenly changed. This aspect should be reviewed throughout the manuscript.
- With respect to the methodology, it would have been interesting to administer more questionnaires -standardized- to correlate the results. This would have helped external validity.
- Regarding the section "4. Variable Measurement and Pilot Test", instead of this, the manuscritp should have a section called Method (and inside the procedure, design and participants) and another section called "Results", separately.
- It would be interesting if all the items of the questionnaire appeared in a table or as an appendix or annex.
- The development of structural equation models is a very good idea and provides the research with robust empirical support.
- The literature should be updated. While using literature from the last five years and even 2019, the proportion of references in that period (at least five years) is low and should be increased. This includes updating the references in the theoretical framework and in the discussion.
In summary, the manuscript analyses the subject matter very well. It would be interesting to update the references to make it more current.
Author Response
Thanks for your comments concerning the manuscript. These comments are very helpful for us to improve our paper. We have revised the manuscript thoroughly, and details can be found in the “track changes” document. In response to your comments and recommendations, the paper has been duly revised and the relevant amendments are summarized as follows:
A1: With respect to the abstract, it would be interesting to introduce the percentage of women or men to get an idea of the representation they have.
Response: Thanks for your comments. We also think that it is necessary and interesting to introduce the proportion of male and female construction professionals in the abstract, which can help readers better understand their representation and the characteristics of the construction industry. We have supplemented the proportion of male and female construction professionals in the abstract (page 1) as follows:
Abstract: This study explored the effects of work-to-family conflict on job burnout and project success in the construction industry. First, a theoretical model with affective commitment as a moderating variable was developed according to the conservation of resources theory. A structured questionnaire survey was then performed with Chinese construction professionals, with 309 valid responses received. In the valid data, the proportion of male construction professionals is 73% and that of female construction professionals is 27%. The analysis of the valid data used structural equation modeling. The results indicate that: (i) work-to-family conflict has a positive and significant effect on job burnout, and a negative and significant effect on project success; (ii) job burnout negatively affects project success; (iii) affective commitment negatively moderates the relationship between work-to-family conflict and job burnout. This study extends the existing body of knowledge on work-to-family conflict and helps to better understand the functional and moderating roles of affective commitment in the context of construction projects. Furthermore, this study provides theoretical guidance and a decision-making reference to help construction enterprises manage work-to-family conflict and job burnout of construction professionals and advance their levels of affective commitment.
A2: The key words could be in alphabetical order and one of them should be related to the research design.
Response: Thanks for your comments. We have arranged the key words in alphabetical order and added the key word "structural equation modeling" related to the research design to the keywords (page 1) as follows:
Keywords: affective commitment; construction professionals; job burnout; project success; structural equation modeling; work-to-family conflict
A3: Both the introduction and the theoretical framework deal in depth with the subject. The hypotheses are highly developed.
Response: Thanks for your very positive comments.
A4: Throughout the manuscript, sometimes the spacing is suddenly changed. This aspect should be reviewed throughout the manuscript.
Response: Thanks for your comments. We have reviewed the spacing of the entire manuscript and revised the problematic spacing.
A5: With respect to the methodology, it would have been interesting to administer more questionnaires -standardized- to correlate the results. This would have helped external validity.
Response: Thanks for your positive comments.
A6: Regarding the section "4. Variable Measurement and Pilot Test", instead of this, the manuscript should have a section called Method (and inside the procedure, design and participants) and another section called "Results", separately.
Response: Thanks for your comments. We have revised the title of the section "4. Variable Measurement and Pilot Test" as "4. Method", the title of the section "4.2. Pilot Test" as "4.2. Pilot Test Procedure", and the title of the section "4.3. Formal Data Collection" as "4.3. Participants and Formal Data Collection". Besides, we have revised the title of the section "4.4. Confirmatory Factor Analysis" as "5. Results of Confirmatory Factor Analysis". After these modifications, this study has a section called "Method" and another section called "Results of Confirmatory Factor Analysis", separately.
A7: It would be interesting if all the items of the questionnaire appeared in a table or as an appendix or annex.
Response: Thanks for your comments. We have put all the items of the questionnaire into the appendix as follows:
Appendix
Title: The questionnaire of "Work-to-Family Conflict, Job Burnout, and Project Success among Construction Professionals: The Moderating Role of Affective Commitment"
Basic information
What is your gender? â–¡Male â–¡female
What is your marital status? â–¡Single â–¡Married
How old are you? â–¡<30 â–¡30–39 â–¡40–50 â–¡>50
Do you have dependent children (aged 18 years or below)? â–¡Yes â–¡No
Do you have elderly dependents? â–¡Yes â–¡No
How long have you worked? â–¡<5 years â–¡6-10 years â–¡11-15 years â–¡16-20 years â–¡>20 years
What is your job position? â–¡Project manager â–¡Department manager â–¡Project engineer â–¡Professional manager â–¡Others
What is your average working hours per week? â–¡<40h â–¡41-50h â–¡51-60h â–¡>60h
Measurements for variables
Work-to-family conflict (WFC)
The requirements of work interfere with my family life.
My work takes too long, so it is difficult for me to perform my family duties.
Because of the demands of my work, what I want to do at home cannot be done.
The pressure of work makes it difficult for me to perform my family duties.
I have no interest in taking part in family activities after work.
Because of my work duties, I have to change my plans for family activities.
Job Burnout (JB)
Work makes me physically and mentally exhausted.
I feel exhausted after work.
When I wake up in the morning and have to face the work of the day, I feel very tired.
It’s stressful for me to work all day.
Work makes me feel like I am falling apart.
Since I started this job, I have become less and less interested in my job.
I'm not as passionate about my work and my colleagues as before.
I doubt the significance of my work.
Affective commitment (AC)
My values are similar to those of the construction enterprise where I work.
I am concerned about the future of the construction enterprise where I work.
I am proud to tell other people that I work in this construction enterprise.
Achieving project goals is as important to me as it is to the project.
I am willing to work harder than ever to help this construction enterprise make progress.
For me, this is the best of all possible construction enterprises for which to work.
Project Success (PS)
The progress of this project is on schedule.
This project is within budget.
The project passed the acceptance check and was successfully delivered.
Most problems encountered in the implementation of the project can be addressed.
The project process is satisfactory.
The owner is satisfied with the project results.
The project meets the special requirements of client.
We look forward to cooperating with the other party again in the future.
A8: The development of structural equation models is a very good idea and provides the research with robust empirical support.
Response: Thanks for your very positive comments.
A9: The literature should be updated. While using literature from the last five years and even 2019, the proportion of references in that period (at least five years) is low and should be increased. This includes updating the references in the theoretical framework and in the discussion.
Response: Thanks for your comments. We have updated the literature and added references from the last five years. This includes updating the references in the theoretical framework and in the discussion. After the update of the references, the proportion of references cited from the past five years has increased.
A10: In summary, the manuscript analyses the subject matter very well. It would be interesting to update the references to make it more current.
Response: Thanks for your very positive comments. We have updated the literature and added references from the last five years. The proportion of references cited from the past five years has increased.
Reviewer 2 Report
Thanks for the opportunity to review this manuscript.
The following comments may help the authors to improve the paper quality.
- Under the theoretical background section, theory support is lacking to support the overall theoretical model.
- In table 3 the authors have reported the model fit values for each construct. CFA is done by including all of the study constructs in the single analysis.
Author Response
Thanks for your comments concerning the manuscript. These comments are very helpful for us to improve our paper. We have revised the manuscript thoroughly, and details can be found in the “track changes” document. In response to your comments and recommendations, the paper has been duly revised and the relevant amendments are summarized as follows:
B1: Under the theoretical background section, theory support is lacking to support the overall theoretical model.
Response: Thanks for your comments. We have supplemented the theoretical framework in the section of “2.4. Project Success” as follows:
2.4. Project Success
Project success has been a hot topic in the field of construction project management [39–42]. Existing research on project success tend to combine project governance with project management and takes the perspective of the whole project life cycle [25]. Furthermore, combining project success and project management success is getting increasingly closer related [39]. However, there has been no consensus on the connotation of project success. This is because different project participants have different benefit demands, resulting in different definitions of project success criteria and the key factors of project success based on different stakeholder perspectives (e.g., owners, contractors, supervisors, consultants, designers) [40]. For example, Shenhar et al. [41] suggested that measurements of project success include project efficiency, effects on customers, business success, and future success. Berssaneti and Carvalho [42] indicated that the project success refers to the iron triangle of time, quality, cost, and customer satisfaction. Luo et al. [4] proposed that project success is defined by time, cost, quality and safety, stakeholder satisfaction, and business value. Therefore, project success can be evaluated from the perspective of stakeholders and the inherent characteristics of construction projects.
This study focus on the effects of work-to-family conflict and the job burnout of construction professionals on project success. Work-to-family conflict and the job burnout of construction professionals mainly occurs during the project implementation; as such, the effect of these factors on project success also occurs at this stage. With respect to construction project stakeholders, although their focus is different, the evaluation criteria for project success during the construction stage are essentially the same, and include hard and soft indicators [25]. Hard indicators include control objectives of quality, cost, duration, and safety [4]. Soft indicators include project management effectiveness, stakeholder satisfaction, future cooperation opportunities, and the level of trust among the stakeholders [25].
The theoretical framework of this study uses the job demands resources (JD-R) model. The JD-R model is a mainstream conceptual framework for studying work stress, work-to-family conflict and job burnout. This model describes the impact of job characteristics on job burnout. According to JD-R model, job characteristics can be classified into two categories: job demands and job resources. Job demands refer to the social or organizational requirements involved in an individual’s work that require continuous energy or time or skills, which in turn is related to certain physical and psychological costs, such as work pressure and work-to-family conflicts. Job resources refer to material, psychological or organizational resources that help individuals achieve their work goals. According to the JD-R model, researchers can better understand and predict individual job burnout. In the construction industry, long working hours and heavy tasks reflect the job demands of construction professionals. These high-load job demands will bring work-to-family conflicts, thus putting construction professionals at high risk of job burnout. Therefore, this study uses the JD-R model for theoretical analysis and develops research hypotheses to investigate the relationship between work-to-family conflict, job burnout and project success among construction professional.
B2: In table 3 the authors have reported the model fit values for each construct. CFA is done by including all of the study constructs in the single analysis.
Response: Thanks for your comments. In our study, the model fitting values of each variable is to test whether the measurement items of each variable reflect the variable. According to your suggestion, we have supplemented the confirmatory factor analysis (CFA) that contains all variables. The results of the supplementary CFA are shown in the following new Table 3.
Table 3. Results of confirmatory factor analysis.
|
Variables |
CR |
AVE |
Fit indices |
|
Work-to-family conflict |
0.87 |
0.79 |
; RMSEA=0.078; GFI=0.94; AGFI=0.93; NFI =0.95; IFI=0.90 |
|
Job burnout |
0.84 |
0.74 |
; RMSEA=0.071; GFI=0.95; AGFI=0.92; NFI =0.94; IFI=0.91 |
|
Project success |
0.81 |
0.71 |
; RMSEA=0.060; GFI=0.93; AGFI=0.91; NFI =0.95; IFI=0.94 |
|
Affective commitment |
0.79 |
0.67 |
; RMSEA=0.067; GFI=0.94; AGFI=0.90; NFI =0.91; IFI=0.92 |
|
All variables |
0.76 |
0.65 |
; RMSEA=0.063; GFI=0.91; AGFI=0.93; NFI =0.91; IFI=0.92 |
Reviewer 3 Report
Dear authors,
Thank you for giving me the opportunity to read and review the ijerph-772579, entitled “Work-to-Family Conflict, Job Burnout, and Project Success among Construction Professionals: The Moderating Role of Affective Commitment”.
I believe that the manuscript is well-written and has potential, but I have some comments that should be considered in order to enhance the manuscript’s quality.
Broad comments
In section 4.3, the authors write “a non-probability sampling method was adopted to obtain a representative sample”, however, a convenience sampling scheme could never produce representative data unless the sample size is extremely large and/or some other assumptions are met.
In section 5, Levene’s test of homogeneity of variances was conducted, however, this is not the appropriate statistic to test for cross-group mean differences.
Figure 4 is misleading because the authors assume a latent variable interaction, but this interaction between affective commitment and work-to-family conflict is not correctly reflected in the path diagram. A revision of the path diagram is probably necessary to preclude any misunderstandings because the way the authors chose to represent the moderating effect of affective commitment references to the classical moderation with manifest dummy variables. Also, as the model is not complex, please include the measurement part of the model along with the residual correlations that the model requires.
Some further clarifications from the authors are of utmost importance regarding their depicted model. Did the authors form cross-product terms of the indicators. If so, which methodology did they follow? Have they included the meanstructure as well? If the authors followed the appropriate approach, they did not mention any information about the procedures they implemented. In-detail description of all the procedures followed must be provided.
In table 4, please provide also all standard errors and critical ratios of the parameters.
Specific comments
In page 9, the term “potential variable” is mentioned but there is no such a thing. Do the authors mean latent variable?
In page 10, the authors mention “validity rate”. Do they mean response rate?
In page 12, the authors state “factor loads”, however, the proper term is “factor loadings”.
In section 7.1, instead of “moderating role” the authors write “mediation role”.
Kind Regards.
Author Response
Thanks for your comments concerning the manuscript. These comments are very helpful for us to improve our paper. We have revised the manuscript thoroughly, and details can be found in the “track changes” document. In response to your comments and recommendations, the paper has been duly revised and the relevant amendments are summarized as follows:
C1: In section 4.3, the authors write “a non-probability sampling method was adopted to obtain a representative sample”, however, a convenience sampling scheme could never produce representative data unless the sample size is extremely large and/or some other assumptions are met.
Response: Thanks for your comments. In this study, a non-probability sampling method was adopted to obtain a representative sample. The use of such a sampling method was considered to be appropriate, because the non-probability sampling is more suitable for this in-depth quantitative study in which we focus on a complex and professional phenomenon in the construction industry. In addition, the respondents were not randomly chosen from the population, but were chosen from professionals in the construction industry and based on their willingness to participate in the study. These measures ensure that the respondents are practitioners in the construction industry and are willing to provide the required information in the questionnaire. Based on the above statement, we have revised the section of “4.3. Participants and Formal Data Collection” as follows:
4.3. Participants and Formal Data Collection
There was no sampling framework for this survey; as such, a non-probability sampling method was adopted to obtain a representative sample [25]. This approach was determined to be appropriate, because the non-probability sampling is more suitable for this in-depth quantitative study in which we focus on a complex and professional phenomenon in the construction industry. In addition, the respondents were not randomly chosen from the population, but were chosen from professionals in the construction industry and based on their willingness to participate in the study [74]. These measures ensure that the respondents are practitioners in the construction industry and are willing to provide the required information in the questionnaire. The survey samples were collected from construction professionals in Shanghai, Zhejiang Province, and Jiangsu Province in China, and included technicians and middle-level and high-level managers of the owner teams, contractor teams, supervisor teams, designer teams, and consultant teams from different construction projects. A total of 1,200 questionnaires were distributed using email and express delivery; 386 questionnaires were returned. After questionnaire screening, 309 were determined to be valid, with a response rate of 26% (309/1200). This was considered normal, based on a 20–30% response rate in most construction industry studies [75]. These valid data were used for reliability and validity analysis and structural equation modeling analysis. Before these analyses, we also applied a Q–Q plot to perform a normality test of the valid data. The results showed that these valid data aligned with a normal distribution. Additionally, the sample structure of valid questionnaires was analyzed; Table 2 shows the categories and levels.
C2: In section 5, Levene’s test of homogeneity of variances was conducted, however, this is not the appropriate statistic to test for cross-group mean differences.
Response: Thanks for your comments. In our study, Levene’s test of homogeneity of variances was considered to be the appropriate method to test for cross-group mean differences. This is because Levene’s test of homogeneity of variances is mainly used to test whether the variances of two or more groups of samples are homogeneous, and this test method only requires the samples to be random. However, other common test methods, such as Bartlett’s test of homogeneity of variances, are mainly used to test the data subject to normal distribution, and its test results for the data subject to deviation of normal distribution are inaccurate. Levene’s test can be used to test the data subject to normal distribution or the data subject to unclear distribution. Besides, the three conversion forms of data in Levene’s test principle ensure the robustness and accuracy of the results. Based on the above statement, we have revised the section of “6. Model Testing and Results” as follows:
6. Model Testing and Results
Before testing the theoretical model, we considered specific demographic variables, including gender and marital status, as these may affect job burnout and project success [80]. Levene’s test of homogeneity of variances was considered to be the appropriate method to test for cross-group mean differences. This is because Levene’s test of homogeneity of variances is mainly used to test whether the variances of two or more groups of samples are homogeneous, and this test method only requires the samples to be random. However, other common test methods, such as Bartlett’s test of homogeneity of variances, are mainly used to test the data subject to normal distribution, and its test results for the data subject to deviation of normal distribution are inaccurate. Levene’s test can be used to test the data subject to normal distribution or the data subject to unclear distribution. Besides, the three conversion forms of data in Levene’s test principle ensure the robustness and accuracy of the results.
SPSS 23.0 was used to examine the influence of these control variables on dependent variables. The results indicate that the effect of gender and marital status on the project success is not significant (gender, −0.127, p > 0.05; marital status, 0.043, p > 0.05). In addition, considering that the work experience of employees and their job position may impact project success [1,81], this study considered these variables to complement related research. The results show that employees’ work experience and job position have no significant impact on project success (work experience, 0.184, p > 0.05; job position, 0.059, p > 0.05). Older and younger construction professionals may respond differently to work-to-family conflict and job burnout [1]. As such, a homogeneity of variance test was performed to assess work-to-family conflict (Levene statistic = 0.218, p > 0.05) and job burnout (Levene statistic = 0.103, p > 0.05). The results show that the homogeneity of variance hypothesis is effective, indicating that the older and younger construction professionals respond similarly to work-to-family conflict and job burnout.
C3: Figure 4 is misleading because the authors assume a latent variable interaction, but this interaction between affective commitment and work-to-family conflict is not correctly reflected in the path diagram. A revision of the path diagram is probably necessary to preclude any misunderstandings because the way the authors chose to represent the moderating effect of affective commitment references to the classical moderation with manifest dummy variables. Also, as the model is not complex, please include the measurement part of the model along with the residual correlations that the model requires.
Response: Thanks for your comments. We referred to previous literatures (e.g., Wu et al., 2019; Haq et al., 2019; Chen et al., 2017; Kim et al., 2009; Lingard and Francis, 2006) related to the test of the moderating effect and confirmed that figure 4 was correct. In order to make readers better understand the moderating effects of affective commitment, a simple slope analysis was implemented. Figure 4 shows the slope. Non-parallel lines represent the existence of moderating effects. The green, blue, and red lines present the moderator’s high, medium, and low conditions, respectively. The moderation analysis indicates that affective commitment negatively moderates the relationship between work-to-family conflict and job burnout, which further verifies the moderating role of affective commitment. In addition, we referred to the previous studies (e.g., Wu et al., 2019; Haq et al., 2019) that also used the method of graphical slope analysis to test the moderating effect. Since the graphical slope analysis shows the test results of the moderating effects in the form of pictures, we did not put the measurement part of the model along with the residual correlations behind figure 4.
References
[1] Wu, G.; Hu, Z.; Zheng, J. Role Stress, Job Burnout, and Job Performance in Construction Project Managers: The Moderating Role of Career Calling. Int. J. Environ. Res. Public Health 2019, 16, 2394.
[2] Lingard, H.; Francis, V. Does a supportive work environment moderate the relationship between work-family conflict and burnout among construction professionals? Constr. Manag. Econ. 2006, 24, 185–196.
[3] Kim, B.P.; Murrmann, S.K.; Lee, G. Moderating effects of gender and organizational level between role stress and job satisfaction among hotel employees. Int. J. Hosp. Manag. 2009, 28, 612–619.
[4] Chen, Y.; Chen, Y.; Liu, Z.; Yao, H. Influence of prior ties on trust in contract enforcement in the construction industry: Moderating role of the shadow of the future. J. Manag. Eng. 2017, 34, 04017064.
[5] Haq, S.U.; Gu, D.; Liang, C.; Abdullah, I. Project governance mechanisms and the performance of software development projects: Moderating role of requirements risk. Int. J. Proj. Manag. 2019, 37, 533–548.
C4: Some further clarifications from the authors are of utmost importance regarding their depicted model. Did the authors form cross-product terms of the indicators. If so, which methodology did they follow? Have they included the mean structure as well? If the authors followed the appropriate approach, they did not mention any information about the procedures they implemented. In-detail description of all the procedures followed must be provided.
Response: Thanks for your comments. This study formed the cross-product terms of the indicators when verifying the moderating effect of affective commitment. The approach we adopted was cross-product terms analysis, which included the mean structure analysis. We have supplemented the test procedures description of the moderating effect of affective commitment in the section of “6. Model Testing and Results” as follows:
This study formed the cross-product terms of the indicators when verifying the moderating effect of affective commitment. The approach we adopted was cross-product terms analysis, which included the mean structure analysis. A simple slope analysis was also implemented to better understand the moderating effect of interaction terms [1]. Figure 4 shows the slope. Non-parallel lines represent the existence of moderating effects. The green, blue, and red lines present the moderator’s high, medium, and low conditions, respectively. The moderation analysis indicates that affective commitment negatively moderates the relationship between work-to-family conflict and job burnout, which further verifies the moderating role of affective commitment.
C5: In table 4, please provide also all standard errors and critical ratios of the parameters.
Response: Thanks for your comments. We have supplemented all standard errors and critical ratios of the parameters in table 4 as follows:
Table 4. Results of theoretical model analysis.
|
Hypothesis |
Path Coefficient |
C.R. Values |
S.E. Values |
T Statistics |
p Values |
Hypotheses Decision |
|
WFC→JB |
0.406*** |
12.124 |
0.193 |
6.624 |
0.000 |
H1: Supported |
|
JB→PS |
-0.347*** |
-4.986 |
0.176 |
-5.428 |
0.000 |
H2: Supported |
|
WFC→PS |
-0.129** |
-3.286 |
0.105 |
-3.246 |
0.001 |
H3: Supported |
|
WFC×AC→JB |
-0.132* |
-2.526 |
0.101 |
-2.513 |
0.014 |
H4: Supported |
|
Fit indices (the full model) |
; RMSEA=0.067; GFI=0.93; AGFI=0.91; NFI =0.92; IFI=0.95 |
|||||
Note: WFC, work-to-family conflict. JB, job burnout. AC, affective commitment. PS, project success. C.R., critical ratio. S.E., standard error. *, p < 0.05. **, p < 0.01. ***, p < 0.001.
C6: In page 9, the term “potential variable” is mentioned but there is no such a thing. Do the authors mean latent variable?
Response: Thanks for your comments. The term “potential variable” in the section of “4.1. Questionnaire Design” (original page 9) means the latent variable. We have revised the term “potential variable” as the term “latent variable” in the section of “4.1. Questionnaire Design”.
C7: In page 10, the authors mention “validity rate”. Do they mean response rate?
Response: Thanks for your comments. The term “validity rate” in the section of “4.2. Pilot Test Procedure” (original page 10) means the response rate. In order to make readers better understand, we have revised the term “validity rate” as the term “response rate” in the section of “4. Method”.
C8: In page 12, the authors state “factor loads”, however, the proper term is “factor loadings”.
Response: Thanks for your comments. We have revised the term “factor loads” as the term “factor loadings” in the section of “5. Results of Confirmatory Factor Analysis” (original page 12).
C9: In section 7.1, instead of “moderating role” the authors write “mediation role”.
Response: Thanks for your comments. We have revised the term “mediation role” as the term “moderating role” in the section of “8.1. Conclusions” (original section 7.1).
Reviewer 4 Report
In general terms, it is a well-constructed and developed manuscript in its different sections. It has a solid conceptual base, especially with regard to the employment situation of construction professionals, which offers an extensive, thorough and in-depth analysis. The hypotheses are adequately supported and the statistical analyzes are rigorous and consistent with the objectives of the study. The discussion and conclusions are also pertinent, in general terms.
Although these evaluations lead me to positively value the study as a whole, I have noticed some slight deficiencies that I understand should be overcome in order to definitively endorse the publication of the work. The following are the deficiencies detected:
Introduction
1) The authors hypothesize the existence of a positive relationship between work-to-family conflict and job burnout in construction professionals, but they do not cite previous studies that support this hypothesis. A simple search through databases reveals the existence of previous research on this relationship, both in construction professionals and in other work contexts. They should be taken as a reference when justifying the hypothesized relationship.
2) The cynicism dimension of burnout must be defined more rigorously. Cynical attitudes develop not only regarding work but, fundamentally, towards human relationships (clients and colleagues).
3) Page 4: "Many job burnout studies have found that stress from work, organization and society in the work context is the main predictor of job burnout [9,10]." In general, burnout is considered to be the result of a conjugation of contextual and personal factors (e.g., low self-efficacy, emotional intelligence, personality traits). This confluence should be reflected in the study.
4) The importance of the variable affective commitment could be justified by taking as a reference the approach based on positive organizational behavior, which has amply demonstrated its positive implications for health and job outcomes. (e.g., Bakker, A. B., & Schaufeli, W. B. (2008). Positive organizational behavior: Engaged employees in flourishing organizations [Editorial]. Journal of Organizational Behavior, 29(2), 147–154. https://doi.org/10.1002/job.515; Luthans, F. (2002). The need for and meaning of positive organizational behavior. Journal of Organizational Behavior, 26, 695–706).
5) The hypothesis of the moderating effect of affective commitment should be better justified. For example, in light of the interaction between contextual and personal factors in the explanation of burnout. Even according to COR theory.
Questionnaire design
1) Why did the authors elaborate a questionnaire having validated instruments for work-to-family conflict, burnout, and affective commitment? In any case, the logical thing would have been to adapt existing questionnaires (with verified evidence of their validity and reliability) to the cultural context.
2) "The first step was to identify measurement items in the existing literature shown to have a high-level of reliability and validity [54]." From which instruments were the items selected? Under what criteria?
Discussion
1) Effects of Work-to-Family Conflict on Project Success
"The results show that work-to-family conflict negatively affects project success. This finding is inconsistent with the conclusion of Allen et al. [81], who found no significant relationship between work-to-family conflict and job outcome." There are numerous studies in other professional contexts that support the positive relationship between work-to-family conflict and poor job outcomes. A more exhaustive search of the literature should be made to cite any of these studies that would be consistent with the results found. Perhaps they would also help to interpret the results differently.
2) The Moderating Role of Affective Commitment: this effect should be interpreted in accordance with COR theory.
3) Practical implications: It would be interesting to interpret the practical implications according to the approach based on sustainable working conditions/positive occupational behavior and its beneficial effects on health and job performance, as evidenced by numerous works in different organizational contexts.
Author Response
Thanks for your comments concerning the manuscript. These comments are very helpful for us to improve our paper. We have revised the manuscript thoroughly, and details can be found in the “track changes” document. In response to your comments and recommendations, the paper has been duly revised and the relevant amendments are summarized as follows:
D1: In general terms, it is a well-constructed and developed manuscript in its different sections. It has a solid conceptual base, especially with regard to the employment situation of construction professionals, which offers an extensive, thorough and in-depth analysis. The hypotheses are adequately supported and the statistical analyzes are rigorous and consistent with the objectives of the study. The discussion and conclusions are also pertinent, in general terms.
Response: Thanks for your very positive comments.
D2: The authors hypothesize the existence of a positive relationship between work-to-family conflict and job burnout in construction professionals, but they do not cite previous studies that support this hypothesis. A simple search through databases reveals the existence of previous research on this relationship, both in construction professionals and in other work contexts. They should be taken as a reference when justifying the hypothesized relationship.
Response: Thanks for your comments. We have supplemented the citation of previous studies that support the positive relationship between work-to-family conflict and job burnout in the section of “3.1.1. Work-to-Family Conflict and Job Burnout” as follows:
3.1.1. Work-to-Family Conflict and Job Burnout
Work-to-family conflict is a source of pressure that makes employees unable to effectively fulfill family responsibilities [6]. This can have negative impacts, such low job satisfaction and high turnover rates [15]. According to the COR theory, people work to acquire and maintain resources that contribute to their objectives, such as improving the quality of their life and increasing family wellbeing [43]. However, when someone is unable to effectively perform their family responsibilities, there is likely to be the potential or actual loss of resources, such as a decreased sense of well-being, poor quality marital relationships, and even divorce [16]. The threat of resource loss is a major cause of pressure and is likely to trigger job burnout [44]. In this case, people may address the pressure by taking measures such as quitting a job to minimize the loss of resources and to protect their remaining resources [6]. Anderson et al. [45] pointed out that work-to-family conflict leads to low work satisfaction and high turnover intention, which may lead to job burnout. Lambert et al. [46] found that work-to-family conflict has a positive relationship with job burnout among correctional staff. Wang et al. [47] revealed that work-to-family conflict has a positive effect on job burnout among female nurses.
In the context of construction projects, heavy workloads, inflexible scheduling, changing requirements, and complex tasks and processes make construction professionals unable to effectively fulfill their family responsibilities [1,6]. This leads to work-to-family conflict, resulting in lost energy and personal time, ultimately causing job burnout. The direct result of job burnout is that construction professionals leave their organization and find another appropriate job to better balance the requirements of work and family domain [6]. Moreover, the loss of resources (e.g., well-being, personal time, and energy) can lead construction professionals to produce negative emotions [48]. These negatively affect their physical and mental health. Therefore, the following hypothesis were proposed:
Hypothesis 1 (H1). Work-to-family conflict positively influences job burnout.
D3: The cynicism dimension of burnout must be defined more rigorously. Cynical attitudes develop not only regarding work but, fundamentally, towards human relationships (clients and colleagues).
Response: Thanks for your comments. We have enriched the definition of the cynicism dimension of job burnout in the section of “2.2. Job Burnout” as follows:
2.2. Job Burnout
Job burnout is a chronic emotional fatigue, caused by constant and daily job stress [9]. It is generally believed that job burnout includes three dimensions: emotional exhaustion, cynicism, and low professional efficacy [30]. Emotional exhaustion involves the feeling that emotional resources are exhausted, leading to a lack of vitality. Cynicism is characterized by a cynical attitude and an exaggerated distancing from one’s work and human relationships, such as with colleagues and clients. Low professional efficacy involves professionals’ negative evaluation of themselves and their dissatisfaction with their work achievements.
D4: Page 4: "Many job burnout studies have found that stress from work, organization and society in the work context is the main predictor of job burnout [9,10]." In general, burnout is considered to be the result of a conjugation of contextual and personal factors (e.g., low self-efficacy, emotional intelligence, personality traits). This confluence should be reflected in the study.
Response: Thanks for your comments. We have added the antecedents of job burnout—personal factors to the analysis of job burnout and revised the section of “2.2. Job Burnout” as follows:
Job burnout is considered to be the result of a conjugation of contextual factors (e.g., heavy workloads, long working hours, great pressure) and personal factors (e.g., low self-efficacy, emotional intelligence, personality traits). However, many job burnout studies have found that stress from work, organization and society in the work context is the main predictor of job burnout in the construction industry [9,10]. In the context of construction projects, due to the temporary, one-time, and uncertain nature of construction projects, professionals face many challenging tasks and unforeseen project situations during the project implementation [6,25]. Construction professionals also have important responsibilities for the cost, duration, quality, and safety objectives of a construction project [2]. As a result, they experience significant stress from work, organization, and society over a long period, from the start of a project to delivery. This causes a high risk of job burnout. Existing research has investigated job burnout among construction professionals, and the results have shown that the level of job burnout and emotional exhaustion of construction professionals are significantly greater compared to workers on other categories of projects [8–11]. Therefore, job burnout is a serious problem in the construction industry.
D5: The importance of the variable affective commitment could be justified by taking as a reference the approach based on positive organizational behavior, which has amply demonstrated its positive implications for health and job outcomes. (e.g., Bakker, A. B., & Schaufeli, W. B. (2008). Positive organizational behavior: Engaged employees in flourishing organizations [Editorial]. Journal of Organizational Behavior, 29(2), 147–154. https://doi.org/10.1002/job.515; Luthans, F. (2002). The need for and meaning of positive organizational behavior. Journal of Organizational Behavior, 26, 695–706).
Response: Thanks for your advices. The term “positive organizational behavior” is novel and interesting, and the book and papers you recommend are valuable. We will further explore the importance of the variable affective commitment and the issues related to affective commitment, work-to-family conflicts and job outcomes in future research.
D6: The hypothesis of the moderating effect of affective commitment should be better justified. For example, in light of the interaction between contextual and personal factors in the explanation of burnout. Even according to COR theory.
Response: Thanks for your comments. We have supplemented the interpretation of the moderating effects of affective commitment according to the interaction between contextual and personal factors and COR theory in the section of “3.1.4. The Moderating Effects of Affective Commitment” as follows:
3.1.4. The Moderating Effects of Affective Commitment
Work-to-conflict is a source of role pressure, and job burnout is considered a direct pressure response [6,8]. Thus, there is a strong pressure-strain relationship between work-to-family conflict and job burnout. However, affective commitment may alleviate this relationship, because it can reduce the urge to save resources by employees who experience work-to-family conflicts. Affective commitment reflects the relationship between employees and their organizations [18]. Employees with affective commitment are featured by an attachment to their organizations, their recognition of the objectives of the organization, pride in their organizations, and their strong desire to remain in the organization [36]. Therefore, employees with affective commitment have a strong sense of ownership and regard the interests of the organization as their own. They are also willing to invest more effort to realize organizational objectives, even if many required actions go beyond their role responsibilities [55]. According to the COR theory, individuals who are highly loyal and attached to the organization do not think it is a frustrating thing to work long hours or have constant work pressure. They also do not reduce their job satisfaction and conserve their own resources such as time and energy because of the high workload. Instead, they are more willing to devote their time and energy to contribute to the organization.
Construction professionals work long and irregular hours under immerse stress, ultimately leading to work-to-family conflict [16]. Construction professionals who are stressed due to work’s interfering with family life tend to attribute such pressure to high-intensity work [56]. This can result in low job satisfaction and trigger job burnout. In this case, construction professionals with affective commitment can modify the attribution level and remain satisfied [34]. Furthermore, they are more likely to reduce the urge to save resources, thereby reducing job burnout [20,21]. As a result, the present study highlights that affective commitment can, to some extent, reduce the job burnout caused by work-to-conflict. Hence, the following hypothesis were proposed:
Hypothesis 4 (H4). Affective commitment can negatively moderate the relationship between work-to-family conflict and job burnout.
D7: Why did the authors elaborate a questionnaire having validated instruments for work-to-family conflict, burnout, and affective commitment? In any case, the logical thing would have been to adapt existing questionnaires (with verified evidence of their validity and reliability) to the cultural context.
Response: Thanks for your comments. In this study, we have elaborated a questionnaire to collect valid data on work-to-family conflict, job burnout, affective commitment, and project success of professionals in the construction industry. The design of all measurement items in the questionnaire were designed with reference to previous studies and combined with the characteristics of the construction industry. In addition, the measurement scales of work-to-family conflict, job burnout, affective commitment and project success have been proved to be applicable in the context of Chinese culture and the Chinese construction industry in our previous studies. Our previous research is as follows:
[1] Wu, G.; Liu, C.; Zhao, X.; Zuo, J. Investigating the relationship between communication-conflict interaction and project success among construction project teams. Int. J. Proj. Manag. 2017, 35, 1466–1482.
[2] Wu, G.; Liu, C.; Zhao, X.; Zuo, J.; Zheng, J. Effects of fairness perceptions on conflicts and project performance in Chinese megaprojects. Int. J. Constr. Manag. 2019, 1–17.
[3] Cao, J.; Liu, C.; Wu, G.; Zhao, X.; Jiang, Z. Work–Family Conflict and Job Outcomes for Construction Professionals: The Mediating Role of Affective Organizational Commitment. Int. J. Environ. Res. Public Health 2020, 17, 1443.
[4] Wu, G.; Wu, Y.; Li, H.; Dan, C. Job Burnout, Work-Family Conflict and Project Performance for Construction Professionals: The Moderating Role of Organizational Support. Int. J. Environ. Res. Public Health 2018, 15, 2869.
D8: "The first step was to identify measurement items in the existing literature shown to have a high-level of reliability and validity [54]." From which instruments were the items selected? Under what criteria?
Response: Thanks for your comments. In this study, all items were designed based on relevant previous studies. The items used to measure work-to-family conflict were designed according to previous studies (Liu and Low [14]; Carlson et al. [62]; Netemeyer et al. [63]). The items used to measure job burnout were designed with reference to the relevant literature (Lingard and Francis [64]; Yip and Rowlinson [65]). The items used to measure affective commitment were also designed according to previous studies (Mowday et al. [37]; Meyer et al. [66]). The items used to measure project success were designed with reference to the relevant literature (Joslin and Müller [25]; Jiang et al. [67]; Pinto et al. [68]). The selection criteria of the measurement items are determined according to the test results of the reliability and validity of the items in each variable measurement scale in previous studies. The reliability coefficients of corrected-item total correlation (CITC) and Cronbach’s α were used to explore the reliability and validity of the items in each variable measurement scale. CITC reflected the reliability of the items. Items with CITC value greater than 0.5 were selected. Cronbach’s α reflected the validity of the items, which should not be below 0.7. We have revised the section of “4.1. Questionnaire Design” as follows:
4.1. Questionnaire Design
To test the theoretical model, a questionnaire was designed to measure the studied variables. Basic demographic data such as family information were also investigated. Study variables included work-to-family conflict, job burnout, affective commitment, and project success. Three steps were applied to develop the questionnaire’s measurement scale. The first step was to identify measurement items in the existing literature shown to have a high-level of reliability and validity [57]. The original scales were developed in English; as such, all items were back-translated and modified [58]. The second step was to modify and improve the existing measurement items according to the characteristics of Chinese construction projects [14]. The third step was to optimize the measurement items by conducting on-site discussions with experts in construction project management [25].
All items were designed based on relevant previous studies [1,6]. The selection criteria of the measurement items are determined according to the test results of the reliability and validity of the items in each variable measurement scale in previous studies. The reliability coefficients of corrected-item total correlation (CITC) and Cronbach’s α were used to explore the reliability and validity of the items in each variable measurement scale [4]. CITC reflected the reliability of the items. Items with CITC value greater than 0.5 were selected [25]. Cronbach’s α reflected the validity of the items, which should not be below 0.7.
Work-to-family conflict was measured using 6 items; job burnout was measured using 8 items; affective commitment was measured using 6 items; and project success was measured using 8 items. The measurement model used in this study provided the relationships between work-to-family conflict, affective commitment, job burnout, and project success (latent variables), and their respective groupings (observable variables) [59]. Furthermore, this study was consistent with the reflection model, because each observable variable on measurements reflected the latent variables [60]. The relationship directions were from the latent variables to the observable variables [61]. Eventually, the items used to measure work-to-family conflict were designed according to previous studies [14,62,63]. The items used to measure job burnout were designed with reference to the relevant literature [64,65]. The items used to measure affective commitment were also designed according to previous studies [37,66]. The items used to measure project success were designed with reference to the relevant literature [25,67,68].
D9: "The results show that work-to-family conflict negatively affects project success. This finding is inconsistent with the conclusion of Allen et al. [81], who found no significant relationship between work-to-family conflict and job outcome." There are numerous studies in other professional contexts that support the positive relationship between work-to-family conflict and poor job outcomes. A more exhaustive search of the literature should be made to cite any of these studies that would be consistent with the results found. Perhaps they would also help to interpret the results differently.
Response: Thanks for your comments. We have conducted a more exhaustive search of the literature, and cited studies that are consistent with the results that work-to-family conflict negatively affects job-related outcomes. We have revised the section of “7.3. Effects of Work-to-Family Conflict on Project Success” as follows:
7.3. Effects of Work-to-Family Conflict on Project Success
The results show that work-to-family conflict negatively affects project success. This finding is consistent with the finding of An et al. [84] and Hao et al. [85], who found a significant negative relationship between work-to-family conflict and job outcome in China. However, this finding is inconsistent with the conclusion of Allen et al. [86], who found no significant relationship between work-to-family conflict and job outcome in America. This may be because as China’s economic and social development and the quality of people’s lives gradually improve, the pursuit of people has shifted from simply pursuing higher income to focusing on individual health and high-quality professional welfare [14]. High-quality professional welfare is closely and significantly related to individual well-being and job satisfaction [6]. In the setting of construction projects, due to the characteristics of long and irregular working hours, constant changing project requirements, and complex tasks and processes, construction professionals must invest significant time and energy at work, leading to work-to-family conflict [1,14]. The significant psychological pressure brought by work-to-family conflict leads to negative emotions. These are likely to cause a low level of job satisfaction and well-being, ultimately increasing turnover. The rapid economic development of China has provided construction professionals with more employment opportunities and higher prospects for rotation than ever before [6]. As a result, many construction professionals quit their jobs to seek another one with improved professional welfare. These negatively impact the completion of project tasks, ultimately negatively affecting project success.
D10: The Moderating Role of Affective Commitment: this effect should be interpreted in accordance with COR theory.
Response: Thanks for your comments. We have supplemented the interpretation of the moderating effects of affective commitment according to the COR theory in the section of “7.4. The Moderating Role of Affective Commitment” as follows:
7.4. The Moderating Role of Affective Commitment
The results show that the affective commitment of construction professionals negatively moderates the relationship between work-to-family conflict and job burnout. This means that the higher the affective commitment of construction professionals is, the weaker the negative impacts of work-to-family conflict on job burnout is. This finding extends our understanding of the functional role and moderating role of affective commitment in the setting of construction projects. It also complements the existing body of knowledge associated with work-to-family conflict in the construction domain, by exploring how affective commitment alleviates the relationship between work-to-family conflict and job burnout.
Following the arguments of self-justification, construction professionals who suffer from great psychological stress from work-to-family conflict tend to attribute this pressure to high-intensity work [56]. This results in their low job satisfaction and eventually trigger their job burnout. However, affective commitment can alleviate this pressure-strain relationship between work-to-family conflict and job burnout. As a driving force, affective commitment is expressed by employees’ emotional attachment to their organization, adherence to the organization’s values and objectives, and a strong willingness to invest effort to achieve the organization’s goals [36]. Therefore, construction professionals with affective commitment can modify their level of attribution and remain satisfied. According to the COR theory, employees who are attached to the organization, proud of the organization, and identified with the organization's goals and values will not be frustrated and anxious about the continuous pressure and high load at work. They are more willing to contribute their own resources such as time and energy to the organization rather than conserve their own resources. Therefore, employees with affective commitment will not reduce job satisfaction due to continuous pressure and heavy tasks at work, which will reduce their likelihood of experiencing job burnout.
D11: Practical implications: It would be interesting to interpret the practical implications according to the approach based on sustainable working conditions/positive occupational behavior and its beneficial effects on health and job performance, as evidenced by numerous works in different organizational contexts.
Response: Thanks for your advices. In our follow-up studies, we will try to interpret the practical implications according to the approach based on the effects of sustainable working conditions or positive occupational behavior on health and job performance.